# Antigenic Targets for the Immunotherapy of Acute Myeloid Leukaemia

**DOI:** 10.3390/jcm8020134

**Published:** 2019-01-23

**Authors:** Ghazala Naz Khan, Kim Orchard, Barbara-ann Guinn

**Affiliations:** 1Department of Biomedical Sciences, University of Hull, Hull HU7 6RX, UK; G.Khan@hull.ac.uk; 2Department of Haematology, University Hospital Southampton NHS Foundation Trust, Southampton SO16 6YD, UK; kho@soton.ac.uk

**Keywords:** Acute myeloid leukaemia, cancer-testis antigen, human, clinical trial, immunotherapy

## Abstract

One of the most promising approaches to preventing relapse is the stimulation of the body’s own immune system to kill residual cancer cells after conventional therapy has destroyed the bulk of the tumour. In acute myeloid leukaemia (AML), the high frequency with which patients achieve first remission, and the diffuse nature of the disease throughout the periphery, makes immunotherapy particularly appealing following induction and consolidation therapy, using chemotherapy, and where possible stem cell transplantation. Immunotherapy could be used to remove residual disease, including leukaemic stem cells from the farthest recesses of the body, reducing, if not eliminating, the prospect of relapse. The identification of novel antigens that exist at disease presentation and can act as targets for immunotherapy have also proved useful in helping us to gain a better understand of the biology that belies AML. It appears that there is an additional function of leukaemia associated antigens as biomarkers of disease state and survival. Here, we discuss these findings.

## 1. Introduction

Acute Myeloid Leukaemia (AML) is rare in children, but is more commonly observed in adults over the age of 65. For context, in the United Kingdom (UK) there were 3126 new cases of AML in 2015 and 2601 deaths from AML in 2016, in a population of 65 million. AML incidence has increased more than 30% since the 1990s and the mortality rate has increased more than 79% since the early 1970s (https://www.cancerresearchuk.org/health-professional/cancer-statistics/statistics-by-cancer-type/leukaemia-aml/incidence) [1]. This likely reflects the ageing population and prior exposure to treatments for cancer, radiation, benzene, and pre-conditions, such as Down Syndrome (www.nhs.uk/conditions.acute-myeloid-leukaemia) [2]. Typically, at diagnoses, the bone marrow sample comprises of about 1 × 10^12^ blast cells and prognosis depends on the severity of the illness at the point of diagnosis. Patients with AML usually present with complications of disordered haematopoiesis: bleeding, fatigue, refractory infections, or the clinical consequences of an extremely high white blood cell count: difficulty breathing, confusion, or other symptoms of organ failure [3]. We have been interested in identifying the antigens that are expressed by AML cells for three reasons. They can (i) act as targets for immunotherapy, (ii) provide new information about the biology of the disease, and (iii) act as biomarkers for the best treatment options or survival.

Immunotherapy stimulates the body’s own immune system to recognise and kill cancer cells and potentially protect against cancer development in the future. It is known that one of the functions of the immune system is to prevent tumour growth, and this is exemplified by the increased tumour frequencies seen in immunocompromised patients following organ transplantation, those with acquired immune deficiency syndrome (AIDS), and in patients with severe combined immunodeficiency (SCID) syndrome [4]. A range of immunotherapy strategies that engage the innate and more often the adaptive immune system have been developed to treat AML (recently reviewed in [5]).

Survival for patients with AML has the potential to be greatly impacted by immunotherapy. Similar to all leukaemias, AML rapidly spreads throughout the body making localised treatments used for solid tumours, such as radiotherapy, of no real benefit. In addition, almost all AML patients will achieve first remission where minimal residual disease (MRD) can be monitored in anticipation of an all too frequent relapse. Around 70–80% of AML patients that were aged less than 65 achieve remission through chemotherapy treatment [6], but around half relapse in the absence of stem cell transplantation (SCT). During this period the immune system can recover and residual disease in difficult to reach places could be eliminated by immunotherapy. Indeed, we already use immunotherapy to treat AML patients through allo-SCT [7]. To boost this anti-tumour response, patients are given donor leukocyte infusions (DLIs) as follow-up treatments post-transplant to maximise the chances of the transplant being successful. Even with SCT, over one-third of patients will relapse [8], and we know that the mortality rates that are associated with SCT, though decreased with the advent of peripheral blood (PB) based haematopoietic-SCT (HSCT), still remain high. Indeed, patients are often exempted from SCT due to a lack of a suitable donor or because they are too fragile to cope with the rigours of SCT, although reduced intensity regimens have made SCT available to a broader base of older patients [9].

We already know a lot about how the immune system works from transplantation studies for AML patients, especially around the importance of graft-versus-host disease (GvHD) to achieve graft-versus-leukaemia (GvL) through twin studies and T cell depletions [10], the boosting of GvHD through repeated DLI transfusions [11] and the role of reduced intensity conditioning allo-transplants to improve the outcomes for older patients [12]. However many patients, especially those who are ineligible to have a HSCT transplant, will relapse after first remission and require further chemotherapeutic treatments [13]. Ideally, patients could be treated with immunotherapy in first remission, to delay or hopefully prevent relapse. 

Currently, the median survival for AML is around one year; however, there has been a steady increase in the overall survival in younger patients [14]. The shift from bone marrow SCTs to PB SCTs has increased donor availability and MRD allows for the prediction of relapse and prophylactic care. However, to date, the largest improvements in survival remain due to improvements in palliative and supportive care [3]. 

## 2. Immunotherapy

Although conventional treatments can be successful for patients with leukaemia, with five-year survival rates for those patients treated with conventional chemotherapeutics (e.g., cytarabine and daunorubicin), being at 27.4% (National Cancer Institute, https://seer.cancer.gov/statfacts/html/amyl.html) [15] in comparison to those who were treated with SCTs being at 44.1%, at five-years post-diagnosis [16]. The success of SCTs needs to be considered in a background of 15–25% mortality [17], due to the treatment itself. On the whole aggressive types and stages are still particularly challenging to diagnose and treat. The future of cancer treatment is increasingly focussed on immunotherapy [18] used in combination with conventional treatments, which is seen as the best opportunity for personalised and more effective treatments that could significantly increase survival rates [19], and in the case of liquid tumours, could remove residual disease at diffuse sites in the body.

The ideal immunotherapy targets should play a role in tumour progression [20], so that tumour destruction targets those cells that are responsible for the tumours aggression as well as starting a cascade of activation induced cell death (AICD), immune stimulation in the context of ‘danger’ and inflammation, and epitope spreading. To ensure monies from National Institute of Health (NIH) grants prioritised immunotherapeutic treatments that focussed on a limited number of antigenic targets, maximising the speed with which treatments reached clinical trials, Cheever and colleagues [21] identified 75 cancer antigens and evaluated them based on nine characteristics that were identified as being essential for effective treatment. p53 [22] was identified as one of the most desirable targets for immunotherapy as targeting p53 can kill both the evolving tumour cell population and any cancer “stem” cell that harbours this as an early stage aberration. By targeting p53, you prevent its support of further tumour growth and genomic instability [23]. However p53, like many other antigens is found to be expressed in solid tumours, but is absent or expressed at low frequencies in haematological malignancies [24]. Indeed, of the antigens considered, those that have been found with any frequency in AML were limited to Wilms’ Tumour protein (WT1) (3rd out of 75) and survivin (12th out of 75), reflecting the authors’ need to provide a shortlist of antigens relevant to as many solid and haematological malignancies as possible. However, the re-expression of some of the antigens listed has been demonstrated through demethylation agents, such as 5′aza-2′-deoxy-cytidine, in recent studies, including, but not limited to, melanoma antigen (MAGE)-A3 (29th of 75), NY-ESO-1 (34th of 75) [25], and synovial sarcoma X breakpoint 2 (SSX2) (53rd of 75) [26].

## 3. The Role of Immunotherapy to Prevent or Delay Relapse in AML Patients in Remission

Treatment for leukaemia is often successful and a first remission achieved [27] however, recurrence is seen in about 50% of younger patients and 90% of older patients [28]. MRD monitoring can predict relapse 2–3 months prior to the development of clinical symptoms [29], enabling prophylactic treatment to give patients the best chance of remaining in remission. The death of patients with leukaemia are generally due to disease relapse and patients in first complete remission who are positive for MRD prior to SCT were more likely to die (2.61 times) or relapse (4.9 times) a second time than patients who were MRD negative [30]. 

Immunotherapy provides an opportunity to remove MRD from cancer patients in first remission, when the burden of disease is low and their immune system is recovering from induction and consolidation therapies. In addition, immunotherapy can be specific to the diseased cells, unlike chemotherapy [31], and destroy leukaemic blast cells in the PB and organs throughout the body. There are a number of different types of antigens [32], including differentiation, mutated, overexpressed, and cancer-testis antigens (CTAs), some of which have been found in AML, including antigens from mutated genes such as Nucleophosmin 1 (NPM1), DNA methyltansferase 3A (DNMT3A), Fms Related Tyrosine Kinase 3 (FLT3), and Ten–Eleven Translocation 2 (TET2) (recently reviewed by [33]). The CTAs category includes some of the oldest and best characterized families, and although MAGE family members were not found to be expressed in presentation AML patient samples with any notable frequency [34], helicase antigen (HAGE) and Per ARNT SIM domain containing 1 (PASD1) antigens have been [34,35]. The differentiation antigens category is another large group of molecules that includes, among many others, the well-known Carcinoembryonic antigen (CEA), glycoprotein 100 (gp100), melan A/melanoma antigen recognized by T cells (MART-1), prostate specific antigen (PSA), and tyrosinase antigens, but relatively few AML antigens have come from this category. The myeloid differentiation antigen CD65 is found at low levels in the least differentiated forms of AML (M0, M1), and usually appears as CD34 disappears during normal myeloid development, reflecting the lack of differentiation in the blast cells in these disease states. The largest group are the overexpressed antigens that include human epidermal growth factor receptor 2 (ErbB-2), human telomerase reverse transcriptase (hTERT), Mucin1 (MUC1), mesothelin, PSA, prostate specific membrane antigen (PSMA), survivin, WT1, p53 and cyclin B1, some of whom are discussed below.

## 4. CTAs

We are particularly interested in CTAs, whose expression is usually restricted to healthy major histocompatability complex (MHC) class I-deficient germline cells (reviewed by [32]). This feature makes them appealing targets for immunotherapeutic strategies because they provide tumour-specific antigens for MHC class I-restricted CD8+ T cells [36]. Developing immunogenic cancer vaccines that target these antigens has become a priority in how cancer is diagnosed and treated. Boon and colleagues were the first to clone a human tumour antigen, named MAGE-1 [37], through the analyses of responses of cytotoxic T cells to melanoma cells. Subsequently, other CTAs were discovered by the group namely the B melanoma antigen (BAGE) and G antigen (GAGE) gene families. Common characteristics of CTAs include mostly being encoded by multigene families, often mapping to the X chromosome and having their expression level epigenetically regulated with drugs, such as 5-aza-2-deoxycytidine [25,26], and although the functions of many are still unidentified, they have been shown to be involved in tumourigenesis [36]. A large number CTAs have been discovered using serological analysis of recombinant cDNA expression libraries (SEREX) [38] showing much promise as biomarkers for disease and providing targets for immunotherapy. Examples include PASD1 in AML [35], LY6K in lung and oesophageal carcinomas [39], sperm protein 17 (Sp17) in head and neck squamous cell carcinoma [40] and transmembrane protein 31 (TMEM31) in metastatic melanoma [41]. The problem is that CTAs are often expressed in less patients (23% for HAGE [34] and 33% for PASD1 [35]) at AML presentation as compared with leukaemia associated antigens (LAAs), such as Survivin [42] and WT1 [43], which are found in most patients and can act as MRD markers in their own right. However CTAs are restricted in their expression to cancer/leukaemia cells and they offer an opportunity to circumvent the initiation of auto-immune responses that could destroy healthy tissues in vulnerable patients. 

It has been increasingly apparent that immunotherapy works best when patients have a healthy immune system and low tumour burden. This is exemplified by the increased cancer incidence observed in patients who have been immune suppressed by Human Immunodeficiency virus (HIV) [44], organ transplantation [45], or cancer treatments, such as radiotherapy and/or chemotherapy [46]. It appears likely that immunotherapy will require use in combination with other treatments, such as hypomethylating agents i.e., SGI-110, a derivative of decitabine [47], which has been shown to lead to the re-expression of MAGE-A and NY-ESO-1 in AML blasts, or more recently treatment in a Phase II clinical trial of AML patients with azacitidine and vorinostat, which led to an increased expression of MAGE, renal cell carcinoma antigen (RAGE), LAGE, SSX2, and taxol resistance associated gene-3 (TRAG3) in blasts, which can be recognised when presented to circulating T cells [48]. In addition, anti-CTLA4 or anti-PD-L1 have been shown to enable the memory of the immune system to recognise tumour antigens (reviewed in [49]).

There has been some suggestion of using CTAs vaccines in a preventative manner at the earliest stages before the cancer advances [50], but predicting which patients are at risk of cancer is often limited to inherited cancers, which account for approximately 5% of all of those affected by cancer and predisposing factors such as exposure to carcinogens that may or may not lead to cancer development.

## 5. CTAs and AML

HAGE is part of the DEAD-box RNA helicases that implies that its function may include RNA metabolism in malignant cells [51]. It has been shown to be expressed in a number of tumour types but not healthy tissues [52]. In 2002, Adams et al. [34] investigated the expression of 10 CTAs in presentation samples from 26 AML and 42 CML. They found little or no expression of MAGE-A1, -A3, -A6, -A12, BAGE, GAGE, LAGE-1, NY-ESO-1 or RAGE. In contrast to previous studies of CTAs in AML, Adams et al. found that HAGE was expressed in 23% of AML patient samples by RT-PCR while it was detected in 14.8% (11/74) AML patients by qPCR analysis by Chen et al. [53]. HAGE has been found to be induced in a dose dependent manner by 5-aza-2′-deoxycytidine [54], a treatment now being used in Phase II clinical trials to overcome T cell exhaustion that is caused by AML blast arginase II activity [48]. 

The PASD1 gene was identified through the immunoscreening of testes cDNA libraries [35,55] using the SEREX technique [56]. A number of investigations have demonstrated PASD1 expression in haematological malignancies, including 4/12 (33%) AML samples [35]. In a cohort of haematological malignancy derived cell lines, the sub-cellular localisation of PASD1, as determined by immunostaining with monoclonal antibodies, was variable [57]. The detection of nuclear staining was not unexpected and it likely reflected the presence of a nuclear localisation signal in the common region of the PASD1-1 and PASD1-2 proteins and the role of PASD1 as a transcription factor [58]. 

Immunogenic T-cell epitopes within PASD1a and PASD1b have proved to be more difficult to identify [59,60]. In AML, Hardwick et al. [60] modified HLA-A*02:01 binding PASD1-specific peptides to generate effective T cell responses. One epitope, Pa14, caused limited expansion in CD8+ T cell numbers from two of three HLA-A*02:01 positive, PASD1-positive AML patient samples. This corresponds with the findings of Rezvani et al. [61], who also found AML T cells have limited capacity to respond to stimulation ex vivo. A 2–3 week limited expansion is the maximum that has been achieved prior to AML T cell death. Reasons for the limited responses may be due to the presence of myeloid suppressor cells in mixed lymphocyte assays [62], interleukin-6 (IL-6) secretion by myeloid leukaemia cells [63], and/or defects in T cell populations in myeloid leukaemia patients [61,64]. However, the stimulation of T cells from a colon cancer patient, by Hardwick et al, led to a substantial increase in the number of Pa14-specific T cells to 13.6% of the CD8+ cell population after four rounds of stimulation, with Pa14-specific IFNγ responses being evidenced [60].

PASD1 expression has not been described in solid tumours although the issues around publishing negative results [65] means that there is little record of which solid tumour have been investigated for PASD1 expression. However the absence of PASD1 expression in solid tumours, including basal cell cancer [66] and ovarian cancer [67], has been published suggesting low expression where it has been described.

## 6. The Role of Tumour Antigens as Biomarkers for Survival

Although tumour antigens were identified for their potential to act as targets for immunotherapy, using the patient immune response for their identification, a number of subsequent studies showed that some, but not all of these antigens could also act as biomarkers [68]. Indeed, despite their known role in cancer initiation and progression, some antigens with elevated expression correlated with improved survival. 

Greiner and Guinn theorised that when leukaemia cells with elevated levels of LAAs are destroyed by chemotherapy the clean-up of the dead/dying cancer cells by the immune system leads to the presentation of antigens in an immunogenic and inflammatory context, leading to improved post-treatment immune responses. In acute promyelocytic leukaemia (APL) patients who harbour the t(15;17) translocation, had a decreased expression of Preferentially Expressed Antigen In Melanoma (PRAME**)** that correlated with a shorter overall survival [69], whereas the typically favourable t(8;21) translocation was associated with a higher level of PRAME in AML M2 patients [70]. Greiner et al. [71] had shown a significant correlation between high G250 mRNA expression levels and a longer overall survival (*p* = 0.022) based on DNA microarray data from 116 AML patients. In addition, the SSX2 interacting protein (SSX2IP) has been found to be a marker of improved survival in AML patients who had no cytogenetic aberrations [72], while also being elevated in patients with t(15;17), associated with poor prognosis until the advent of (treatment) and decreased in patients harbouring the more favourable t(8;21) [73]. Guinn et al. found a positive correlation between the expression of SSX2IP and the poor prognostic indicator FLT-3-ITD (*p* = 0.008, *t* test), but not between SSX2IP and other poor prognostic markers, such as cytogenetic abnormalities associated with poor survival, white cell count, age, sex, or survival [73]. 

However this has not been the case with all antigens. Liberante et al. [74] suggested a ‘Goldilocks’ effect of the relative levels of PRAME expression in terms of its role as a biomarker for survival. It was found that ‘very high’ and ‘very low’ levels of PRAME expression correlated with poor survival. Low levels of PRAME expression may reflect a situation where leukaemia cells are able to escape immune surveillance, while higher levels of PRAME could reflect a higher tumour load and/or the presence of more aberrant leukaemia cells [74]. In addition, elevated survivin expression has been shown to correlate with chemoresistance [42] and poor outcomes [75,76] in AML. This is more commonly the case in solid tumours, where the elevated expression of antigens tends to be associated with a worse clinical outcome, if there is an association. Examples include **survivin** in different solid tumours, including renal cell carcinoma [76] and HAGE in breast cancer [77]. In addition, differences between survival and antigen expression can vary with AML subtype, patient age, and cytogenetics perhaps reflecting the heterogeneity of AML. For example, RAGE-1 and MGEA6 were both found to have elevated expression in the less lineage restricted forms of AML [78], while microarray analysis showed elevated SSX2IP in patients with the t(15;17) and significantly decreased levels of SSX2IP in patients harbouring the t(8;21) [73]. 

Bergmann et al. showed that high levels of WT1 mRNA in AML were associated with poor long-term outcome [79], while others found no correlation [71,80,81]. However Bergmann’s findings reflected the situation in non-small cell lung cancer, where low WT1 mRNA expression has been associated with poor survival and lymph node metastases [82]. This may demonstrate the need to further sub-group patients based on age or other demographics. Indeed, the expression of BCL-2 and WT1 has been associated with a reduced rate of achieving complete remission and overall survival in patients that were younger than 60 years, and no effect on survival rates in patients older than 60 years [83]. 

## 7. Antigens that Have Been Shown to Play a Role in the Biological Basis of AML

A number of proteins were identified by virtue of an antibody response against them and were then shown to have an important role in the biological basis of AML. Greiner discussed the role of a number of LAAs in cell cycle proliferation (BAGE, BCL-2, OFA-iLRP, FLT3-ITD, G250, hTERT, PRAME, Hyaluronan-mediated motility receptor (HMMR, also known as RHAMM), proteinase 3, survivin, and WT1), meaning that immunotherapy strategies targeting them would also destroy leukaemic cells that are proliferating abnormally under the control of overexpressed or mutated antigens.

In AML patients with the t(15;17) translocation, SSX2IP levels were associated with gene expression of proteins involved in regulating cyclin dependent kinases (CDK) activity (p57Kip2, cdk7, cyclins D2, D3, E2, and B2), DNA replication (CDC6) and mitosis (survivin and CENPJ) [73]. We also found a very significant correlation between AML patients harbouring a t(8;21) and low cdc20 expression [73]. Boyapati et al. [84] had described a mouse model of AML M2 whose cells had a C-terminal truncated AML-ETO product and developed aneuploidy through the attenuation of the spindle checkpoint. Using microarray datasets for associations between *SSX2IP* and the genes involved in spindle checkpoints described by Boyapati et al., Guinn et al. [73] found a strong correlation between low-*CDC*20 expression, one of the substrate-targeting subunits of the anaphase-promoting complex and low-*SSX2IP* expression in patients harbouring a t(8;21) translocation when compared with AML patients without a t(8;21) translocation and normal donors. 

In 2007, Denniss observed the variable expression of PASD1 in synchronised K562 cells over time [85], but could not demonstrate an association with the phases of the cell cycle. Others also noted that only a subset of K562 cells expressed PASD1 (around 17% of the cell population) [60,86] and they could be reproducibly killed by PASD1-specific T cells [60]. PASD1, a homologue to the mouse CLOCK gene, has now been shown to suppress circadian rhythms. The circadian clock regulates and responds to the physiological and environmental changes by regulating transcription in a roughly 24 h cycle. PASD1 through its interaction with CLOCK:BMAL1 reduces transcription regulation, leading to the transformation of cells. PASD1 C-terminal CC1 domain bears homology to the essential regulatory region encoded by CLOCK exon 19. Using molecular mimicry, PASD1 can restrict the activation of CLOCK exon 19 to disrupt the CLOCK:BMAL1 function, therefore supressing transcription [87].

Survivin, coded by the baculoviral IAP repeat-containing 5 (*BIRC5)* gene, has been shown to be involved in several central pathways that control cell proliferation and viability (reviewed recently by Garg et al. [88]). Of particular note, survivin is a key player of the survivin-Borealin-INCENP core complex that regulates important proteins that are involved in cell division, like aurora B kinase or polo-like kinase 1 [89,90]. Several pathways, such as mTOR- and ran-GTP, are regulated by survivin [91,92], and survivin is involved in spindle formation and anti-apoptosis [91]. While in normal differentiated adult tissues little or no expression of survivin is found, high expression has been described in a number of different solid tumors and hematological malignancies [91]. Attempts to antagonize survivin using antisense molecules are ongoing, including immuno-targeting by vaccination and tyrosine kinase inhibition [93,94,95]. Notably, a repressor of survivin recently produced encouraging results in heavily pretreated cancer patients [96]. 

WT1 has emerged as one of the most promising targets for AML immunotherapy, because of its oncogenic role in leukaemogenesis, its high expression in the majority of AML cells, and its ability to function as a tumour rejection antigen [97]. Concomitantly, many other haematological [98,99,100] and solid [99,100,101] tumours could benefit from WT1-directed therapy. Despite its ubiquitous expression during embryogenesis, WT1 expression in normal individuals is limited to renal podocytes, gonadal cells, and CD34^+^ bone marrow cells [102,103], where expression is significantly lower than in leukaemia cells (10–100 fold) [103], making it an excellent target for immunotherapy. 

## 8. Clinical Trials–State-of-The-Art

As T cells are able to recognise and kill cancer cells [104], it was thought that T cell therapies would be the most effective form of immunotherapy. T cells are believed to have an exquisite specificity for epitopes within tumour antigens and they are able to effectively kill cancer cells in a controlled manner. Cytotoxic T-lymphocytes (CTLs) can be stimulated through the use of dendritic cells (DCs) [105], peptide vaccines [106], DNA vaccines [107], and natural killer (NK) cells [108].

DCs are antigen presenting cells that are able to cross present by ingesting and processing extracellular antigens and presenting them on Major Histocompatability Complex (MHC) class I molecules [109]. DC therapy involves extracting the patient’s own monocytes, maturing and activating them to DCs using antigens. The DCs are then injected back into the body to stimulate the immune system to eliminate the antigen expressing cancer cells [110]. 

AML cell lines were used to show that PRAME is involved in retinoic acid-regulated (RAR) cell proliferation and differentiation by inhibiting RAR signalling [111] and introducing all-trans-retinoic acid (ATRA) may be able to reverse this, especially in patients without the t(15:17) mutation. Combination treatment of targeting PRAME along with ATRA would potentially benefit patients expressing elevated levels of PRAME [111]. The presence of PRAME could be an indicator for relapse, as it was found to be increased, after decreasing during remission, even with multiple relapses [112]. PRAME has been shown to induce specific T-cell responses in both solid tumours and leukaemia [113]. However, in some patients expressing PRAME, the cytotoxic response is too weak but after treatment with a Histone deacetylase (HDAC) inhibitor chidamide enhanced PRAME levels are observed, with further improvement when chidamide is combined with the DNA demethylating agent decitabine resulting in immune cells recognizing the PRAME100–108 or PRAME300–309 peptide presented by HLA-A*02:01 [114]. 

Monoclonal antibodies are used to treat a number of cancers, including low-grade or follicular non-Hodgkin's lymphoma (NHL) and chronic lymphocytic leukaemia (CLL), through treatment with rituximab, which is a CD20 specific antibody. Rituximab targets CD20 that is present on the surface of the B cells, including the malignant NHL and CLL cells [115]. 

The best strategy for the effective treatment of cancer may include a combination of conventional and immunotherapy techniques [116], or even a combination of immunotherapy techniques, as demonstrated in increasing numbers of mouse models [117] and clinical trials [118,119,120]. Subsequently, adoptive T cell therapy has been shown to be very promising with the number of cells being returned to patients [121] and their status–activated but not matured [122], being the main considerations. Chimeric antigen receptors-T cells (CAR-T) are where a patients T cells are genetically engineered to express the CAR receptor on their surface against a specific antigen. Upon expansion, they are injected back into the body to recognise and kill the antigen expressing cancer cells. In a recent novel study, a T-cell receptor-mimic (TCRm) CAR, known as WT1-28z, responded to a peptide portion of the intracellular antigen WT1, as it is presented on the surface of the tumour cell in the context of HLA-A*02:01. T cells genetically modified to recognise WT1-28z specifically targeted and lysed HLA-A*02:01+ WT1+ tumours and improved the survival of mice engrafted with HLA-A*02:01+, WT1+ leukaemia cells [123].

There are a number of excellent reviews in this area of research that aim to identify and discuss effective immunotherapy strategies for the future (Table 1). These include cellular immunotherapy [124], whole cell vaccines [125], multidrug resistance [126], DCs [127], oncolytic viruses [128], and nanotechnology [129]. Targeted therapeutic strategies along with ever improving designs in clinical trials pave the way for further success [130].

In addition, combinations of immunotherapy could further enhance survival, reducing residual disease where there are escape variants. Combining the antibodies anti-CTLA-4 and anti-4-1BB revealed CD8^+^ immune responses against advanced MC38 tumours as well as establishment of memory T cells. Combination treatments reduced autoimmunity in comparison to a single antibody therapy [137] and they often offer an opportunity to eliminate escape variants. Combination therapy could be the answer for drug resistant tumours as the resistance mechanisms of the tumour can be identified and targeted alongside standard treatments. Two cell lines (breast and gastric cancer), resistant to sacituzumab govitecan, became susceptible to therapy through the use of an ATP-binding cassette (ABC) transporter inhibitor that is used in combination with antibody treatment [138]. ABC transporters can cause drug resistance by efflux-removal of the drug from the cell [139]. Promising combination therapies utilising antibodies include Lapatinib with trastuzumab in Her2 positive breast cancer [140], Dabrafenib and Trametinib in relapsed ovarian cancer [141], carboplatin and pemetrexed in advanced non-small cell lung cancer [142], pidilizumab and rituximab in follicular lymphoma [143], albumin-bound paclitaxel and gemcitabine in pancreatic cancer [144], nivolumab and ipilimumab in untreated metastatic melanoma [145], cisplatin and topotecan or cisplatin and gemcitabine in advanced colon cancer [146], and bevacizumab plus oral capecitabine plus irinotecan in metastatic colon cancer [147].

## 9. Summary

We have described the multiplex of insights that novel antigens have provided into how AML develops and how it might be targeted by immunotherapy approaches during disease remission. We have not however discussed novel treatments that we felt were outside the scope of this review and dealt with in detail elsewhere. Obvious examples include CAR-T cells (recently reviewed in [148]), RNA interference (RNAi) targeting, for example, of Brd4 [149], and antibody therapies, including anti-CD33 (recently reviewed in [150])

Poor T cells responses in AML patients [60,61] make gauging anti-tumour responses using ex vivo T cells from AML patients difficult, and expanding immune and leukaemia cells for therapy before patients relapse have struggled to succeed. However, the success of HSCT and DLIs has shown the capacity of the immune system to overcome leukaemia cells when advantaged to do so.

For the monitoring of MRD and effective T cell responses, it is important that proteins specific to the disease are identified and for immunotherapy that cancer specific antigens are the targets of immune responses, including those enacted by B-cell responses (by definition) and their immune counterparts (CD4+ and CD8+ T-cells among others). 

The issues remain when to give vaccines against leukaemia to best impact the disease and the effect of treatment on the immune system cannot be underestimated, especially in myeloid leukaemia. Clinical trials, for what is a relatively rare cancer, as compared to many solid tumours, include a limited number of immunotherapy treatments and perhaps a new list of prioritised tumour antigens for haematological malignancies/leukaemia/myeloid leukaemia are required.

Whatever the way forward for AML treatment, it will undoubtedly require the combination of SCT wherever possible, induction and consolidation therapies to achieve MRD, immune recovery, and a lot of trial and error for this heterogenous population. 

## Figures and Tables

**Table 1 jcm-08-00134-t001:** Some examples of current clinical trials involving antigenic targets in acute myeloid leukaemia (AML).

Target Antigen(s)	Designated Name	Type of Immunotherapy	Phase	Findings	Refs
CD33 and CLL1	CD123b-CD33b cCAR	CAR-T/cellular immunotherapy	I	1 patient–44 year old female. Liu stated that the CD33 cCAR T cell therapy could be used as a conduit to transplant, in addition to conventional chemotherapy or alone.	[131]
MUC1-C plus decitabine	GO-203-2C	Peptide inhibition of MUC1/targeted therapy	I/Ib	Combination cohort, response was achieved in 57% compared to GO-203-2C alone who had resistant disease. Showed treatment is safe.	[132]
Proteinase 3	PR1	Peptide vaccine	I/II	PR1 vaccine induces specific immunity that correlates with clinical response, including molecular remission	[133]
Bcl-2	Venetoclax	Small molecule inhibitor	II	Measurable reduction in bone marrow blast counts was observed in 53% of patients	[134]
WT1	galinpepimut-S	Peptide vaccine	II	Median disease-free survival from CR1 was 16.9 months, whereas the overall survival from diagnosis is estimated to be ≥67.6 months	[135]
hTERT	AST-VAC1	hTERT expressing autologous DCs	II	58% developed T-cell responses, 58% patients in CR were free of relapse after 52 months, 57% of patients aged ≥60 also were free of relapse after 54 months	[136]

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
