# Peer review of "Antigenic Targets for the Immunotherapy of Acute Myeloid Leukaemia"

_jcm, 2019, doi:10.3390/jcm8020134_

Reviewer 1 Report

In this manuscript, authors reviewed the recent progress in the field of immunotherapy of acute myeloid leukaemia (AML), with particular interest in the antigenic targets for AML. This manuscript is of clinical interest and well written. The data is well presented and convincing. There are some concerns.

Major comments:

1.     This review talks about immunotherapy. However, authors even did not gave a definition of immunotherapy, and what kinds of immunotherapy are available in treating AML.

2.     Line 39-40: “Similar to all leukaemias the disease rapidly spreads throughout the body making localised treatments such as radiotherapy of little benefit”. There is misunderstanding. Radiotherapy normally doesn’t apply to leukaemias' treatment.

3.     Line 85-87: “targeting p53 can kill both the evolving tumour cell population and any cancer “stem” cell which harbours this as an early tumourigenesis stage aberration. By targeting p53 you prevent its support of further tumour growth and genomic instability” Please show the references for this statement.

4.     Line 25, should be “in 2016” not “the 2016”.

Author Response

In Response to Reviewer 1 Comments:-

In this manuscript, authors reviewed the recent progress in the field of immunotherapy of acute myeloid leukaemia (AML), with particular interest in the antigenic targets for AML. This manuscript is of clinical interest and well written. The data is well presented and convincing. There are some concerns.

Major comments:

1.     This review talks about immunotherapy. However, authors even did not gave a definition of immunotherapy, and what kinds of immunotherapy are available in treating AML.

The following paragraph defining immunotherapy has been added to page 4 of the review as follows:-

Immunotherapy stimulates the body’s own immune system to recognise and kill cancer cells and potentially protect against cancer development in the future. It is known that the immune system is to prevent tumour growth, and this is exemplified by the increased tumour frequencies seen in immunocompromised patients following organ transplantation, human immunodeficiency virus (HIV) infection and in patients with severe combined immunodeficiency (SCID) syndrome [6]. A range of immunotherapy strategies that engage the innate and more often the adaptive immune system have been developed to treat acute myeloid leukaemia (recently reviewed in [7]).

2.     Line 39-40: “Similar to all leukaemias the disease rapidly spreads throughout the body making localised treatments such as radiotherapy of little benefit”. There is misunderstanding. Radiotherapy normally doesn’t apply to leukaemias' treatment.

This has been corrected on page 3 as follows:-

Similar to all leukaemias, AML rapidly spreads throughout the body making localised treatments used for solid tumours such as radiotherapy of no real benefit.

3.     Line 85-87: “targeting p53 can kill both the evolving tumour cell population and any cancer “stem” cell which harbours this as an early tumourigenesis stage aberration. By targeting p53 you prevent its support of further tumour growth and genomic instability” Please show the references for this statement.

The appropriate reference, Bykov et al, 2018, has been inserted into this line on page 5 of the manuscript as follows ‘By targeting p53 you prevent its support of further tumour growth and genomic instability [20].’

4.     Line 25, should be “in 2016” not “the 2016”

Corrected

Reviewer 2 Report

Peer review on jcm 424220

General:

This is a well-written and comprehensive review of tumor antigens in AML with regard to their role in tumor biology, residual disease monitoring, and potential tumor immunotherapy.  Although the authors state that the field of CAR-T therapy is outside their scope, the reader is expecting from the title at least an outlook on the identification and selection of antigens in AML that might be suitable for a CAR-T concept. This field should therefore be covered at least in brief. Additional examples of CAR-T in AML should be included in table 1.

Minor:

line 100: any treatment initiated by an MRD test result should be termed „preemptive“ and not „prophylactic“

lines 100-103 The sentences “AML patients with cytogenetics abnormalities such as t(8;21) and t(15;17) are 80 % more likely to relapse than those with normal karyotypes in their blasts [24] and patients with the translocation t(8;21) were found to relapse more than once while patients with the t(15;17) translocation showed only one relapse [25]” are misleading, since the mentioned translocations are associated with favourable prognosis AML with a high probability of achieving and maintaining CR. Reference 24 also does not seem to be appropriate to substantiate the given statement. The sentences should be deleted or rewritten.

Line 142: the abbreviation “LAA” is not explained (leukemia associated antigen?)

Line 252: Ref.# 69 needs to be formatted in the reference list

Line 331: the table should additionally explain the type of therapy in a separate column (e.g. “CAR-T / cellular immunothreapy” for ref # 141; “pharmacological inhibitor of bcl-2”, and so on. Are vaccine studies, antibody or antibody-drug conjugates included?

Author Response

In Response to Reviewer 2 Comments:-

General:

This is a well-written and comprehensive review of tumor antigens in AML with regard to their role in tumor biology, residual disease monitoring, and potential tumor immunotherapy.  Although the authors state that the field of CAR-T therapy is outside their scope, the reader is expecting from the title at least an outlook on the identification and selection of antigens in AML that might be suitable for a CAR-T concept. This field should therefore be covered at least in brief. Additional examples of CAR-T in AML should be included in table 1. 

CAR-T cells are discussed on page 14 of the review as follows:-

‘Chimeric antigen receptors T cells (CAR-T) are where a patients T cells are genetically engineered to express the CAR receptor on their surface against a specific antigen. Upon expansion they are injected back into the body to recognise and kill the antigen expressing cancer cells. In a recent novel study, a T-cell receptor-mimic (TCRm) CAR, known as WT1-28z, responded to a peptide portion of the intracellular antigen WT1, as it is expressed on the surface of the tumour cell in the context of HLA-A*02:01. T cells genetically modified to express WT1-28z specifically targeted and lysed HLA-A*02:01+ WT1+ tumours and improved the survival of mice engrafted with HLA-A*02:01+, WT1+ leukaemia cells [122].’

And referred to in the summary paragraph 1 (page 15) with a reference to a recent review

Minor:

line 100: any treatment initiated by an MRD test result should be termed „preemptive“ and not „prophylactic“

Corrected

lines 100-103 The sentences “AML patients with cytogenetics abnormalities such as t(8;21) and t(15;17) are 80 % more likely to relapse than those with normal karyotypes in their blasts [24] and patients with the translocation t(8;21) were found to relapse more than once while patients with the t(15;17) translocation showed only one relapse [25]” are misleading, since the mentioned translocations are associated with favourable prognosis AML with a high probability of achieving and maintaining CR. Reference 24 also does not seem to be appropriate to substantiate the given statement. The sentences should be deleted or rewritten.

These lines have been removed to avoid confusion and reference 24 has been deleted.

Line 142: the abbreviation “LAA” is not explained (leukemia associated antigen?)

Abbreviation has now been explained

Line 252: Ref.# 69 needs to be formatted in the reference list

Completed

Line 331: the table should additionally explain the type of therapy in a separate column (e.g. “CAR-T / cellular immunothreapy” for ref # 141; “pharmacological inhibitor of bcl-2”, and so on. Are vaccine studies, antibody or antibody-drug conjugates included?

The column heading of Type has been changed to Type of immunotherapy as suggested and details added as follows:-

.